# Choosing the best route: Comparative optimization of wheat transformation methods for improving yield by targeting *TaARE1-D* with CRISPR/Cas9

Mumin Ibrahim Tek[1,2,3], Kubra Budak Tek[1], Pelin Sarikaya[4], Abdul Razak Ahmed[4], Hakan Fidan[4]*

1 Département de Phytologie, Université Laval, Québec City, Québec, Canada, 2 Institut de Biologie Intégrative et des Systèmes (IBIS), Université Laval, Québec City, Québec, Canada, 3 ATG biotek, Izmir, Türkiye, 4 Plant Protection Department, Akdeniz University, Antalya, Türkiye

* hakanfidan@akdeniz.edu.tr

## Abstract

Wheat (*Triticum aestivum* L.) is one of the most important crops worldwide, supplying a major share of calories and protein for the global population. Incorporating gene editing into breeding programs is critical to improve yield and stress tolerance, yet wheat remains difficult to transform and regenerate efficiently. These bottlenecks limit the full application of CRISPR/Cas9 for improvement yield in wheat. To address this, transformation parameters were optimized for three methods: immature embryo transformation, callus transformation, and injection-based *in planta* transformation. Systematic optimization of *Agrobacterium* strain, bacterial density, acetosyringone concentration, and incubation conditions resulted in substantially improved transformation success. Efficiencies of 66.84% for immature embryos, 55.44% for callus, and 33.33% for *in planta* transformation were achieved, representing more than tenfold increase compared with previously reported rate of ~3%. A key innovation was the shortening of the callus induction stage for immature embryos, reducing the time required for plant regeneration by approximately one month while maintaining high transformation efficiency. The protocols were validated through CRISPR/Cas9-mediated knockout of *TaARE1-D*, a negative regulator of nitrogen uptake and yield. Generated mutants exhibited increased grain number, spike length, grain length, and thousand-grain weight, as well as the characteristic stay-green phenotype associated with loss of *TaARE1-D* function. The optimized protocols provide robust platforms to accelerate gene-editing in wheat to increase yield and stress-tolerance.

## Introduction

Wheat (*Triticum aestivum* L.) is one of the most important staple crops worldwide, providing a substantial share of the global caloric and protein supply [1]. With the

**Data availability statement:** All relevant data are within the manuscript and its Supporting information files.

**Funding:** The author(s) received no specific funding for this work.

**Competing interests:** The authors have declared that no competing interests exist.

rapidly growing global population and the increasing challenges posed by climate change, there is an urgent need to improve wheat productivity, resilience, and nutritional value. Recent advances in gene-editing technologies, particularly CRISPR/Cas9, have transformed plant breeding by enabling precise and targeted modifications of the genome [2]. This precision facilitates the efficient introduction of desirable traits such as higher yield, enhanced disease resistance, increased tolerance to abiotic stresses, and improved nutritional quality [3,4].

To date, many studies have focused on improving wheat yield by either characterizing yield-associated genes or targeting previously identified negative regulators [2]. For example, *Grain Weight* (*TaGW*) genes, which act as negative regulators of yield, have been frequently targeted, resulting in considerable increases in grain size, grain width, and overall yield [5,6]. Another strategy is modifying spike structure-related traits using nucleases such as Cas9. The loss of function of *DUO-B1*, which encodes an APETALA2/ethylene response factor (AP2/ERF), was shown to increase the number of spikelets and grains per plant [7]. In addition to morphological traits, targeting physiological regulators has also proven effective. A well-known example is the *ARE1* gene, which encodes a regulator of nitrogen uptake in rice. Loss-of-function mutations in *ARE1*, known as *abnormal cytokinin response1 repressor-1* mutants, improved nitrogen use efficiency [8]. Similarly, targeting the wheat ortholog *TaARE1* using CRISPR/Cas9 in the hexaploid AABBDD genome increased nitrogen uptake, leading to longer spikes, delayed senescence, and enhanced grain weight, particularly in *AABBdd* and *aabbDD* mutants [9].

Despite the immense potential of gene editing in wheat, its widespread application remains constrained by major bottlenecks *in planta* transformation and regeneration. Wheat is considered one of the most recalcitrant cereals to genetic transformation, and both the delivery of foreign DNA into cells and the regeneration of whole plants continue to pose significant challenges [2,10]. Among these, the regeneration step is the most limiting, as it is highly genotype-dependent, time-consuming, and labor-intensive [11,12]. These obstacles hinder the efficient introduction of gene-editing components into elite wheat cultivars. Therefore, overcoming transformation and regeneration barriers is essential for integrating CRISPR/Cas9 and related technologies in breeding programs for wheat [7].

Several methods have been developed for wheat transformation, each with distinct advantages and limitations. The most widely used approach involves the transformation of immature embryos, typically through *Agrobacterium*-mediated gene transfer or particle bombardment [13–15]. However, these methods are often genotype-dependent, labor-intensive, and rely on specialized tissue culture facilities. *In planta* transformation strategies, which aim to bypass tissue culture by directly targeting germline cells, have also been investigated [16]. However, these approaches have generally been hindered by low transformation efficiencies and poor reproducibility, limiting their suitability for large-scale breeding programs.

Therefore, in this study, we optimized existing *Agrobacterium*-mediated transformation methods to significantly improve efficiency and reduce the time required to obtain transgenic wheat plants by carefully adjusting key parameters. Using

CRISPR/Cas9, we targeted *TaARE1-D*, a well-characterized negative regulator of nitrogen uptake and yield. In addition, we introduced an improved, faster, and more efficient *in planta* transformation method, providing a promising alternative to conventional approaches. At the same time, we developed accelerated regeneration protocols for both immature and callus explants, achieving efficiencies comparable to traditional but more time-consuming methods. Finally, we comprehensively compared all optimized approaches in terms of transformation efficiency, regeneration rate, gene-editing efficiency, and the overall time from initial transformation to the recovery of edited mature plants with the desired phenotype.

## Materials and methods

### Plant material

Spring wheat (*Triticum aestivum* cv. Kayra), obtained from the Aegean Agricultural Research Institute (AARI, Türkiye), was used as plant material. Plants were cultivated in controlled growth chambers at 24°C under a 16/8-hour light/dark photoperiod and 70% relative humidity. Water and fertilizer were applied regularly to support growth until spikelet formation and anthesis, at which stage immature embryo explants were collected.

### Vector construction

We used the modular cloning system for the CRISPR/Cas9 construct [17,18] to target *TaARE1-D* in the hexaploid genome of *T. aestivum* (S1 Fig). The *TaARE1-A* (TraesCS7A03G0678600), *TaARE1-B* (TraesCS7B03G0545300), and *TaARE1-D* (TraesCS7D03G0650400) sequences were obtained from Phytozome and used to select *TaARE1*-D-specific gRNAs (S1 Table). After Level 1 cloning of the gRNAs, we used the P3 TaU6 guide acceptor (Addgene, 165599) and the P4 TaU6 guide acceptor (Addgene, 165600) in the final assembly with 100 ng of pGoldenGreenGate-M (Addgene, 165422), OsActinP:Hygint:NosT (Addgene, 165423), the pICH41780 linker (Addgene, 48019), and OsUbiP:Cas9:NosT (Addgene, 16542). Each cloning step was performed according to the previously described protocol, after which the plasmids were transferred into *E. coli* DH5α and subjected to blue-white screening [19]. The final assembly was confirmed by *BglII* restriction digestion, PCR, and Sanger sequencing (S2 Fig).

### *Agrobacterium* strain, OD, and acetosyringone optimization

The CRISPR vector was introduced into *Agrobacterium tumefaciens* strains AGL1, EHA105, and GV3101 (GoldBio), all carrying the pSOUP helper plasmid, by electroporation [20]. Transformed colonies were selected on LB agar supplemented with kanamycin (50 µg/mL), rifampicin (25 µg/mL), and tetracycline (5 µg/mL) (S2 Fig). After 2–3 days of incubation, single colonies were transferred into liquid LB medium containing the same antibiotics and incubated at 28°C. Glycerol stocks (40%) were prepared and stored at –82°C for long-term use. For transformation experiments, cells were first cultured in antibiotic-free LB medium (5 mL) for 16–18 hours. On the day of inoculation, cultures were scaled up in 40 mL of fresh LB medium and incubated until an $OD_{600}$ of 1.0 was reached. Cells were pelleted by centrifugation (4,000 rpm) and resuspended in inoculation medium (2.2 g/L MS salts, 10 g/L glucose) supplemented with 100, 150, or 200 µM acetosyringone (Phytotech, A104). Suspensions were maintained at 28°C with shaking (100 rpm) for at least two hours before transformation.

### Transformation of callus derived from mature embryos and regeneration

Mature seeds were sterilized in 70% ethanol for 90 seconds, rinsed three times with sterile distilled water, and treated with 1% commercial bleach for 20 minutes. After three additional rinses, seeds were soaked overnight in sterile water to facilitate embryo extraction. Callus induction was performed on MS-based medium modified according to previous protocol [21]. The callus induction (CI) medium consisted of 4.4 g/L MS salts with vitamins (M519, Phytotech), 30 g/L maltose

(Sigma, M9171), 1.25 mg/L CuSO$_4$ (Phytotech, C375), 1 g/L casein hydrolysate (Sigma, 22090), 2.5 mg/L 2,4-D (Phytotech, D295), and 3.5 g/L Phytagel (Sigma, P8169). Excised mature embryos were placed on CI medium and incubated in the dark at 24°C for 4–6 weeks. The medium was refreshed biweekly. Well-developed calli (6–10 mm diameter) were selected for transformation. For inoculation, 10–15 calli were immersed in 15 mL of *Agrobacterium* suspension and gently shaken for 10, 15, or 20 minutes. Excess bacterial suspension was removed by blotting on sterile filter paper. Then, calli were transferred onto co-culture medium containing 4.4 g/L MS salts with vitamins, 30 g/L maltose, 1 g/L casein hydrolysate, 1.25 mg/L CuSO$_4$, 100, 150, or 200 µM acetosyringone (Phytotech, A104), and 3.5 g/L Phytagel, and incubated in the dark at 24°C for three days. Following co-cultivation, calli were transferred to selection medium (SM) composed of 4.4 g/L MS salts with vitamins, 30 g/L maltose, 1 g/L casein hydrolysate, 1.25 mg/L CuSO$_4$, 15 mg/L hygromycin (Phytotech, H397), 200 mg/L Timentin (Phytotech, T869), either 1 mg/L zeatin (Phytotech, Z899) or 0.2 mg/L indole-3-acetic acid (IAA) (Phytotech, I885), and 3.5 g/L Phytagel. Cultures were maintained at 24°C under a 16/8-hour photoperiod (500–550 µmol m$^{-2}$ s$^{-1}$) and 70% humidity for 2–4 weeks, and calli were monitored daily for shoot formation. Negative transformants were eliminated by hygromycin selection. Regenerated shoots were transferred to rooting medium consisting of 2.2 g/L MS salts with vitamins, 30 g/L maltose, 200 mg/L Timentin, and 3 g/L Phytagel. Healthy plantlets were subsequently acclimatized in pots containing a 3:1 mixture of compost soil and perlite. Pots were covered with transparent plastic bags to maintain high humidity for one week, after which holes (1 cm diameter) were gradually introduced. Bags were removed completely after an additional week. Regenerated plants were subjected to transgene screening, and T1 seeds were harvested from mature plants.

## Transformation of immature and mature embryos and regeneration

Immature and mature embryos were isolated following seed sterilization as previously described [13,21]. Isolated embryos were first placed in liquid inoculation medium containing 0.05% Silwet-L77 (Phytotech, S7777) without *Agrobacterium*. Samples were centrifuged for 10 minutes at 10,000 rpm, after which the medium was replaced with *Agrobacterium* suspension. Embryos were incubated for 10, 15, or 20 minutes, with tubes inverted three to four times every five minutes to ensure uniform exposure. Excess inoculum was removed by blotting on three layers of sterile filter paper. Embryos were then transferred to co-cultivation medium and incubated in the dark for three days before being placed on CI medium supplemented with 200 mg/L Timentin for seven days. Calli with diameters of 4–6 mm were subsequently transferred to CI medium containing 15 mg/L hygromycin and 200 mg/L Timentin and maintained in the dark at 24°C for two weeks. Surviving calli were transferred to SM supplemented with either 0.2 mg/L IAA or 1 mg/L zeatin and incubated at 24°C under a 16/8-hour photoperiod (500–550 µmol m$^{-2}$ s$^{-1}$) and 70% humidity for 2–4 weeks to evaluate shoot regeneration. Healthy shoots that survived selection were subsequently transferred to rooting medium and later acclimatized under greenhouse conditions as described above.

## *in planta* transformation

The *in planta* transformation method has been adapted and optimized from the previously indicated methods [22,23]. Fifteen to twenty sterilized seeds were placed on pre-wetted paper towels in Petri dishes and incubated in the dark at 24°C for three days. After germination, an *Agrobacterium* suspension was delivered using a 1-mL syringe fitted with a 26-gauge needle (0.4 mm × 13 mm). The needle was inserted into the apical meristem to a depth of approximately 0.1 mm, and 10–15 µL of inoculum was applied directly to the wounded tissue. Following injection, seedlings were maintained on wet paper towels in Petri dishes with the lids partially open for two, three, or four days in the dark to ensure high humidity. Explants were then washed with 200 mg/L Timentin for 20 minutes on a shaker (150 rpm, 24°C) and rinsed thoroughly with sterile distilled water. Treated seedlings were transplanted into pots and fertilized gradually. Two weeks after transformation, newly developed leaves were collected for PCR screening. Plants identified as transgenic were advanced for growth and harvested to obtain T1 seeds.

## PCR screening and gene-editing efficiency

Genomic DNA was isolated using the CTAB method [24] from plantlets 2–4 weeks after acclimatization (*in vitro* transformation) or sowing (*in planta* transformation). Transgene screening was performed by PCR in a 20 μL reaction containing 10 μL 2×EcoTaq Master Mix (Ecotech, ET-5), 1 μL each of HygF/HygR primers [13]. 2 μL DNA template, and 7 μL nuclease-free water. Amplification conditions were: initial denaturation at 94°C for 10 min, followed by 35 cycles of 95°C for 35 s, 58°C for 45 s, and 72°C for 45 s, with a final extension at 72°C for 10 min. PCR products were loaded on 1.5% agarose gels, and transgene-positive plants were identified by the presence of the expected band. Negative plants were discarded, while positive plants were maintained under a 16/8-h photoperiod at 24°C in controlled growth chambers until seed harvesting. Gene-editing efficiency was assessed using the ACT-PCR method that allows the rapid detection of homozygous mutants- [25]. PCR primers specific to *TaARE1* in the D genome of wheat were designed after aligning each copy of the gene. The same strategy was followed to design sequencing primers 50–60 bp up- and downstream of the *TaARE*-1D target region. Reaction mixtures and cycling conditions were identical to the screening PCR, except that the annealing temperature was set to 62°C (S1 Fig; S1 Table). Absence of a PCR product was interpreted as a putative mutation caused by disruption of the primer binding site at the Cas9 cleavage site in *TaARE1-D*. Furthermore, DNAs from T0 and T1 plants were used for PCR with TaARE1D-SEQ-F/R primers. The products were subjected to sequencing using the Sanger method. This method was employed to detect not only the types of induced mutations, such as insertions, deletions, and substitutions, but also to confirm the heritability of the induced mutations.

## Phenotyping and data analysis

Edited knockout lines were selected based on the effects of the induced mutations on the TaARE1-D protein sequence (S9 Table). The mutations were interpreted by comparing the amino acid sequences with that of the reference TaARE1-D protein (*TraesCS7D03G0650400*). Only plants carrying frameshift mutations or premature stop codons were advanced for phenotypic analysis.Phenotyping was performed for agronomic traits including number of grains per spike, spike length, grain length, and thousand-grain weight. Measurements were recorded at maturity using a minimum of three biological replicates per genotype including non-edited plants. Statistical analyses were performed in R (v4.5.1). One-way ANOVA was used to compare mean trait values, followed by Tukey HSD for post hoc comparisons. Data visualization, including phenotypic trait plots and optimization parameter outputs, was generated using the ggplot2, dplyr, and tidyr libraries.

## Results

### Optimization of transformation parameters for immature embryo explants

To optimize transformation conditions, different *Agrobacterium* strains were first tested using fixed parameters: $OD_{600} = 0.8$, 150 μM acetosyringone in both inoculum and co-culture media, and an incubation time of 15 minutes. Transformation efficiency was assessed based on the survival of explants following callus formation and selection with 15 mg/L hygromycin (S2 Table). Among the tested strains, AGL1 showed the highest efficiency, with 61.67% of explants surviving after selection, followed by EHA105 at 37.30%, whereas GV3101 showed the lowest survival rate at only 10.42%. Based on these results, AGL1 was chosen for further optimization Fig 1.

Next, we evaluated bacterial density by varying $OD_{600}$. Transformation with $OD_{600} = 0.8$ yielded the highest survival rate (75.64%), compared with 48.90% at 0.6 and 24.52% at 1.0. Incubation time also influenced efficiency: the lowest transformation rate was observed with 10 minutes (27.47%), whereas 15 minutes increased efficiency to 63.95%. Prolonging incubation to 20 minutes reduced survival to 51.10%. Collectively, these findings indicate that the most efficient conditions for immature embryo transformation were obtained using AGL1 at OD600 = 0.8, with a 15-minute incubation, and 150 μM acetosyringone in both inoculum and co-culture media (S2 Table). The same strategy was applied to mature embryos. Although callus formation was observed following transformation, no explants survived under selection. Instead, only necrotic calli developed, and shoots lost viability within one to two days in the selection medium.

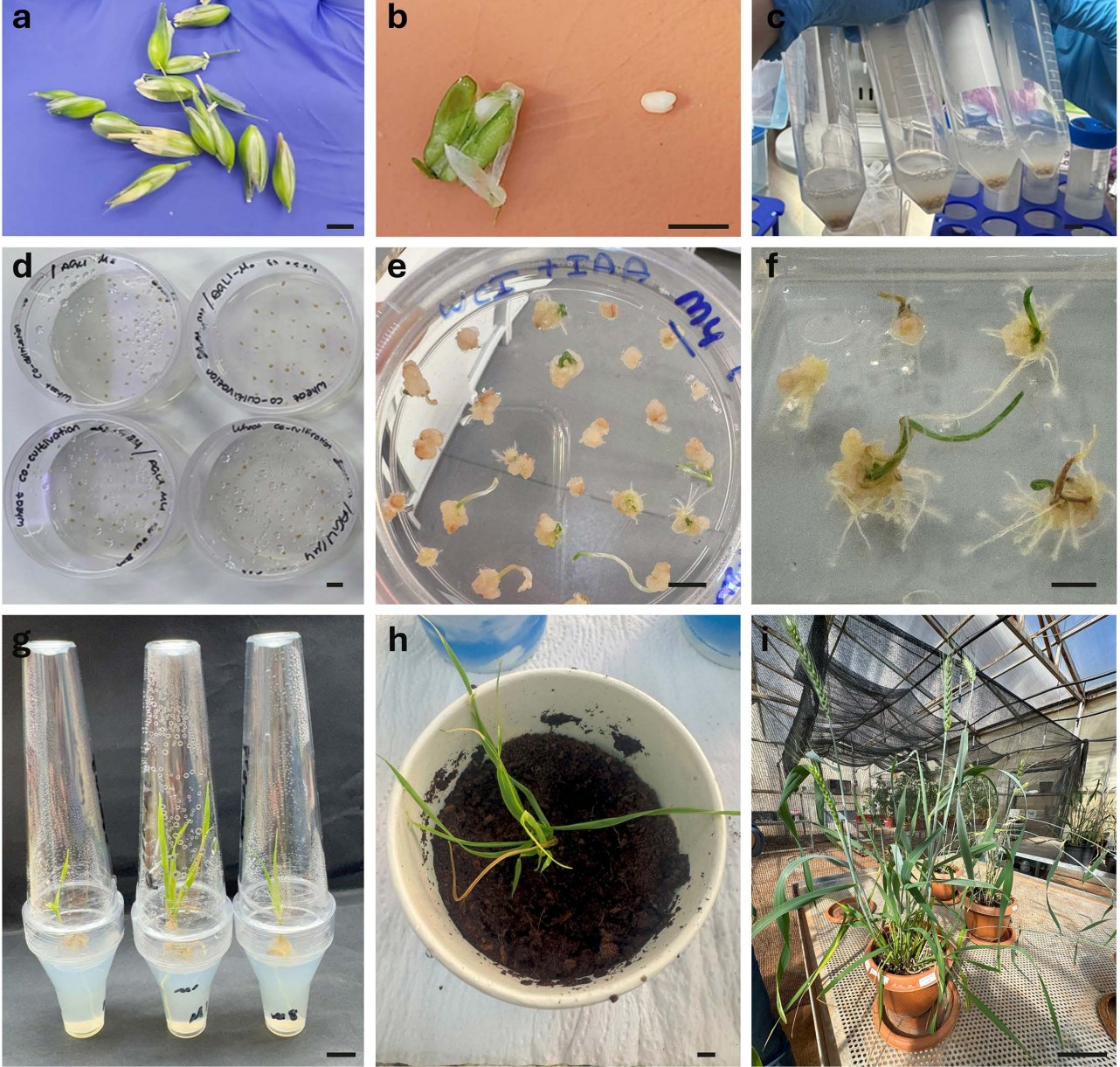

**Fig 1. Transformation and regeneration of wheat plants from immature embryo explants.** a) Spikes were collected from wheat 14-21 days after anthesis. b) Immature embryos were dissected for use in transformation. c) Immature embryos were immersed in *Agrobacterium* for transformation in 50 mL centrifuge tubes. d) Coculture of immature embryos with *Agrobacterium*. e) After one month on callus induction medium (CI), the calli were transferred to selection medium (SM) supplemented with 0.2 mg/L IAA, where shoots and small roots appeared within 3–5 days. f) Surviving shoots were selected on 15 mg/L hygromycin. g) subsequently transferred to rooting medium. h) Regenerated plantlets were acclimatized in soil under high-humidity conditions. i) well-developed plants were maintained until seed harvest. Bar = 0.5 cm in a-b, 1 cm in c-h, and 5 cm in i.

### Optimization of transformation parameters for callus

Callus formation was successfully induced from mature embryos cultured in the dark on CI medium (Fig 2). Well-developed calli began to appear after 2 weeks, reaching 3–4 mm in diameter, and by four weeks they had expanded to 7–8 mm. These calli were collected and subsequently used for transformation experiments. As with immature embryos,

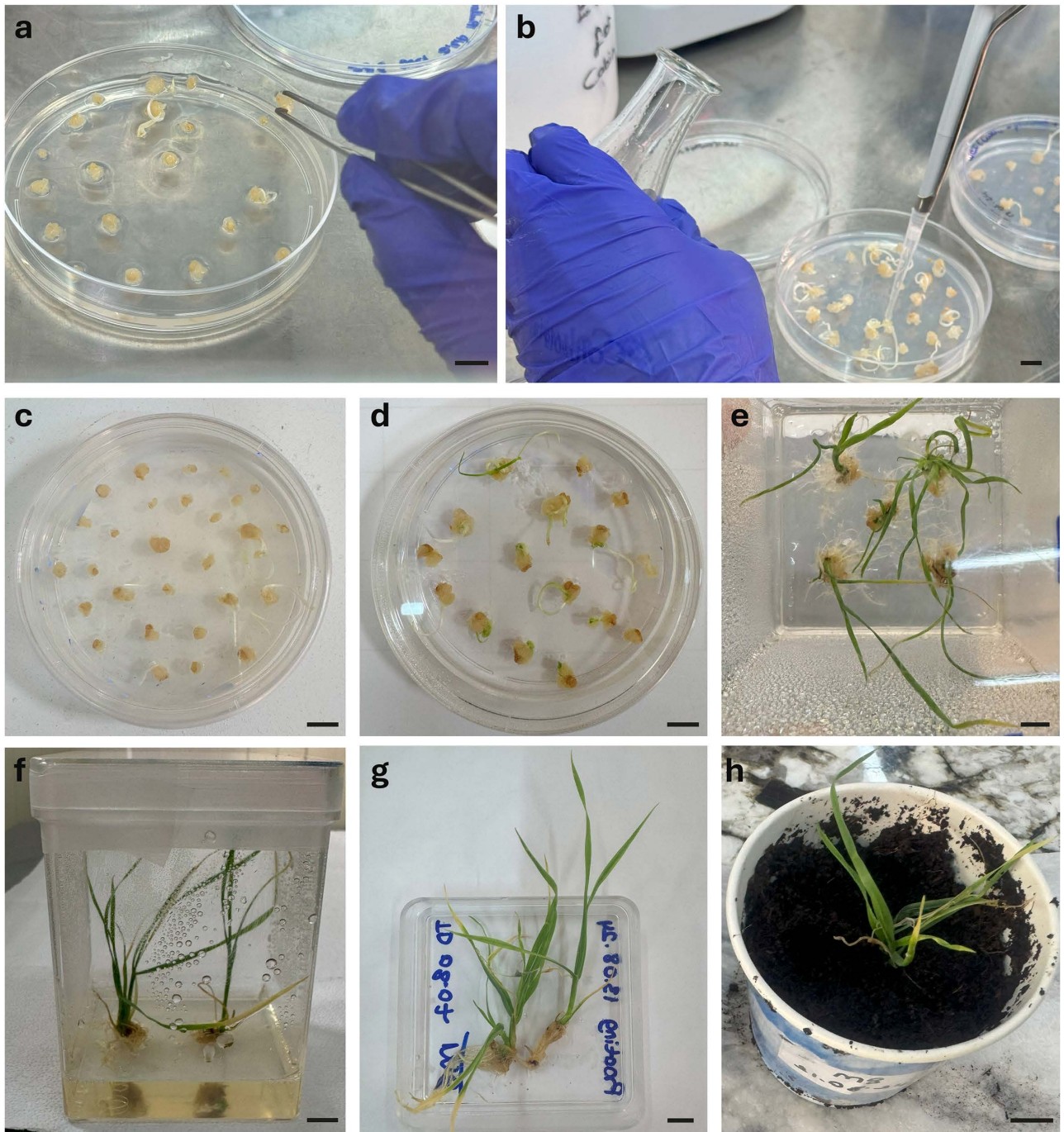

**Fig 2. Transformation and regeneration of wheat plants from callus derived from mature embryos.** a) Callus was induced from mature embryos placed on culture medium. b) Subsequently used for transformation, the callus surface was covered with Agrobacterium suspension during transformation. c) Calli were transferred to selection medium (SM) supplemented with 0.2 mg/L IAA d) First shoots appeared within five days. e) Survived plantlets after four weeks of selection f) Surviving plantlets e were transferred to rooting medium. g) Well-developed plants in rooting medium. h) Acclimatized wheat in soil, they were maintained until seed harvest. Scale bar = 1 cm.

different *Agrobacterium* strains were first tested under fixed parameters: $OD_{600} = 0.8$, 150 µM acetosyringone in inoculum and co-culture media, and a 15-minute incubation. Although overall efficiency was lower than with immature embryo transformation, AGL1 again showed the highest survival rate (51.68%), followed by EHA105 (26.59%) and GV3101 (12.27%). Next, we optimized bacterial density. The highest survival rate was observed at $OD_{600} = 0.8$ (57.51%), while increasing the density to $OD_{600} = 1.0$ reduced survival to 25.71%. No statistically significant difference was observed between OD600 = 0.8 and 0.6, despite a slight difference. The effect of acetosyringone concentration was also examined. While no significant difference was observed between 100 µM and 150 µM, further increasing the concentration substantially reduced survival, with only 12.64% of calli surviving selection. Finally, incubation time was tested. A 10-minute incubation yielded the highest efficiency (73.9%), followed by 15 minutes (63.5%), with no significant difference. However, prolonging the incubation to 20 minutes reduced survival dramatically to 27.68%. In summary, the most efficient conditions for callus transformation were obtained using AGL1 at $OD_{600} = 0.8$, a 10-minute incubation, and 150 µM acetosyringone in the inoculation and co-culture media, with selection on 15 mg/L hygromycin (S3 Table).

## Regeneration of callus

We next tested the effects of IAA (0.2 mg/L) and zeatin (1 mg/L) on the regeneration of wheat calli. After co-cultivation, calli derived from both immature and mature embryos were transferred to selection medium (SM) supplemented with either IAA or zeatin. Shoots regenerated more efficiently on IAA-containing medium than on zeatin-containing medium. To confirm this observation, we evaluated regeneration efficiency in the absence of selection. Explants from immature embryo-derived calli placed on 0.2 mg/L IAA produced shoots within 2–3 days, with a regeneration rate of 52.63%. In contrast, calli placed on 1 mg/L zeatin regenerated at only 26.32% (S4 Table). Similarly, calli derived from mature embryos regenerated at 47.94% on IAA compared with 24.55% on zeatin. Together, these results demonstrate that 0.2 mg/L IAA is substantially more effective than 1 mg/L zeatin for promoting regeneration in both immature and mature embryo-derived calli.

## Improving of *in planta* transformation

Similar to the *in vitro* approach, *in planta* transformation was optimized (Fig 3) by first testing different *Agrobacterium* strains under fixed parameters ($OD_{600} = 0.8$, 150 µM acetosyringone in the inoculum, and three days of dark incubation). Based on PCR screening, AGL1 was again the most efficient strain, yielding 30.42% positive plants. We then tested different cell densities ($OD_{600} = 0.6$, 0.8, and 1.0). Increasing bacterial density improved transformation efficiency, with PCR-positive rates rising from 7.01% at $OD_{600} = 0.6$ to 39.12% at $OD_{600} = 1.0$. Next, acetosyringone concentrations were compared (100, 150, and 200 µM). No significant difference was observed between 100 µM (32.54%) and

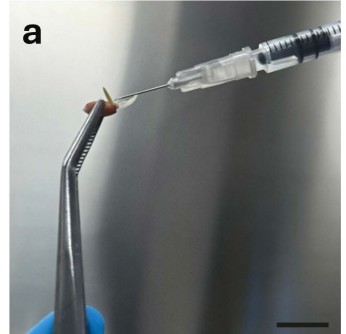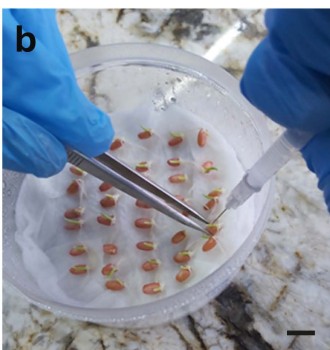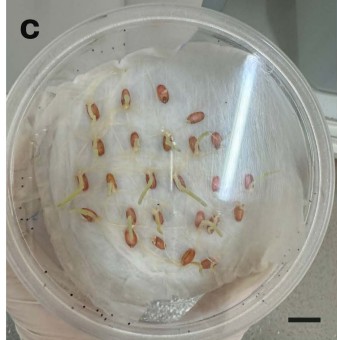

**Fig 3. General workflow of *in planta* transformation.** a, b) Germinated seedlings were injected with Agrobacterium into the plumule part. c) Following transformation, plants were maintained in the dark for at least two days to allow bacterial interaction Bar = 1 cm.

150 µM (36.51%), but efficiency decreased substantially to 15.28% at 200 µM (S5 Table). Finally, we evaluated post-transformation dark incubation. Incubation for two days yielded the highest efficiency (39.73%), whereas extending incubation to three or four days reduced efficiency to 24.52% and 27.14%, respectively. Although there was no significant difference between three and four days, extended dark treatment, particularly four days caused etiolation of seedlings. In summary, the most efficient *in planta* transformation was achieved using AGL1 at OD600 = 1.0, with 150 µM acetosyringone in the inoculum, followed by two days of dark incubation (Fig 3). After treatment, plants were washed with 200 mg/L Timentin for 20 minutes, rinsed three times with distilled water, and transferred to soil for further growth. Interestingly, many *in planta*-transformed plants exhibited over-tillering, with some producing up to 20–25 tillers (S3 Fig).

## Gene-editing efficiency across transformation methods

Using the optimized protocols for each method we evaluated CRISPR/Cas9 gene-editing efficiency. After selection and regeneration, we obtained 43 plants from immature embryo transformation, 40 from callus transformation, and 28 from *in planta* transformation. Transgene screening was performed with HygF/R primers (S4 Fig). Of these, 38 out of 42 immature embryo-derived plants and 30 out of 40 callus-derived plants were PCR positive, whereas 7 out of 28 plants from *in planta* transformation tested positive. We further analyzed randomly selected PCR-positive plants by ACT-PCR to detect mutations induced by CRISPR/Cas9. In total, 22 plants from immature embryos, 15 from callus-derived transformation, and 7 from *in planta* transformation were screened with target-specific primers. Mutation events at the target region were detected in 16 of 22 immature embryo-derived plants, 9 of 16 callus-derived plants, and 3 of 7 *in planta* plants (S5 Fig).

Then, the plants in which the mutation events had occurred were sequenced to detect the type of mutation and confirm the presence of the induced mutations by CRISPR/Cas9. The longest deletion was detected in IM33, obtained through immature embryo transformation, with 8 bp deletion at the gRNA1 target and 4 bp deletion at the gRNA2 target. 2-base-pair deletions were determined at both Cas9 cleavage sites of gRNA1 and gRNA2 in MC7, which was derived from callus transformation. 1-bp deletion was detected at the gRNA2 site in IP5 plants that were transformed in-plant. Furthermore, mutation types were screened in the T1 generation obtained from the transformed and PCR-screened T0s. Most of the mutations were deletions but a 1 bp insertion detected only in IM8, which was obtained via immature embryo transformation. The heritability of the mutation was also confirmed for both methods by sequencing T1 plants derived from the IM33, MC7 and IP5 lines. However, no mutation was found in IP21, despite its T0 being identified as a mutant by PCR.

## Phenotypic analysis of *taare1d* mutants

We compared the *taare1d* mutants with the unedited control variety (*T. aestivum* var. Kayra). Three representative mutant lines; IM33 (immature embryo transformation), MC7 (callus-derived transformation), and IP5 (*in planta* transformation) were evaluated for grain yield-related traits, including number of grains per spike, spike length, grain length, and thousand-grain weight. All three mutant lines showed improved yield parameters compared with the control (Fig 4a). Kayra produced an average of 57.5 ± 2.89 grains per spike, whereas IM33, MC7, and IP5 produced 65 ± 2.89, 64.5 ± 2.69, and 63.2 ± 1.72 grains, respectively (Fig 4b). Spike length also increased, rising from 9.47 cm in Kayra to 10.33 cm in IM33, 10.18 cm in MC7, and 10.20 cm in IP5 (Fig 4c). Grain size and weight were likewise improved. The average grain length increased from 6.69 mm in Kayra to 7.91 mm in IM33, 7.46 mm in MC7, and 7.54 mm in IP5, while thousand-grain weight rose from 40.51 g to 43.77 g, 43.57 g, and 43.09 g, respectively (Fig 4d and 4e). For grain length specifically, MC7 and IP5 grouped statistically with both NE and IM33, reflecting minor differences between these lines.

## Discussion

Wheat is one of the most important staple crops worldwide, providing nearly 20% of global dietary energy and protein intake [26]. Rising global temperatures have already caused an estimated 5.5% reduction in wheat yields between 1980 and 2010 [27]. Consequently, the development of new wheat varieties with higher yield potential and improved tolerance

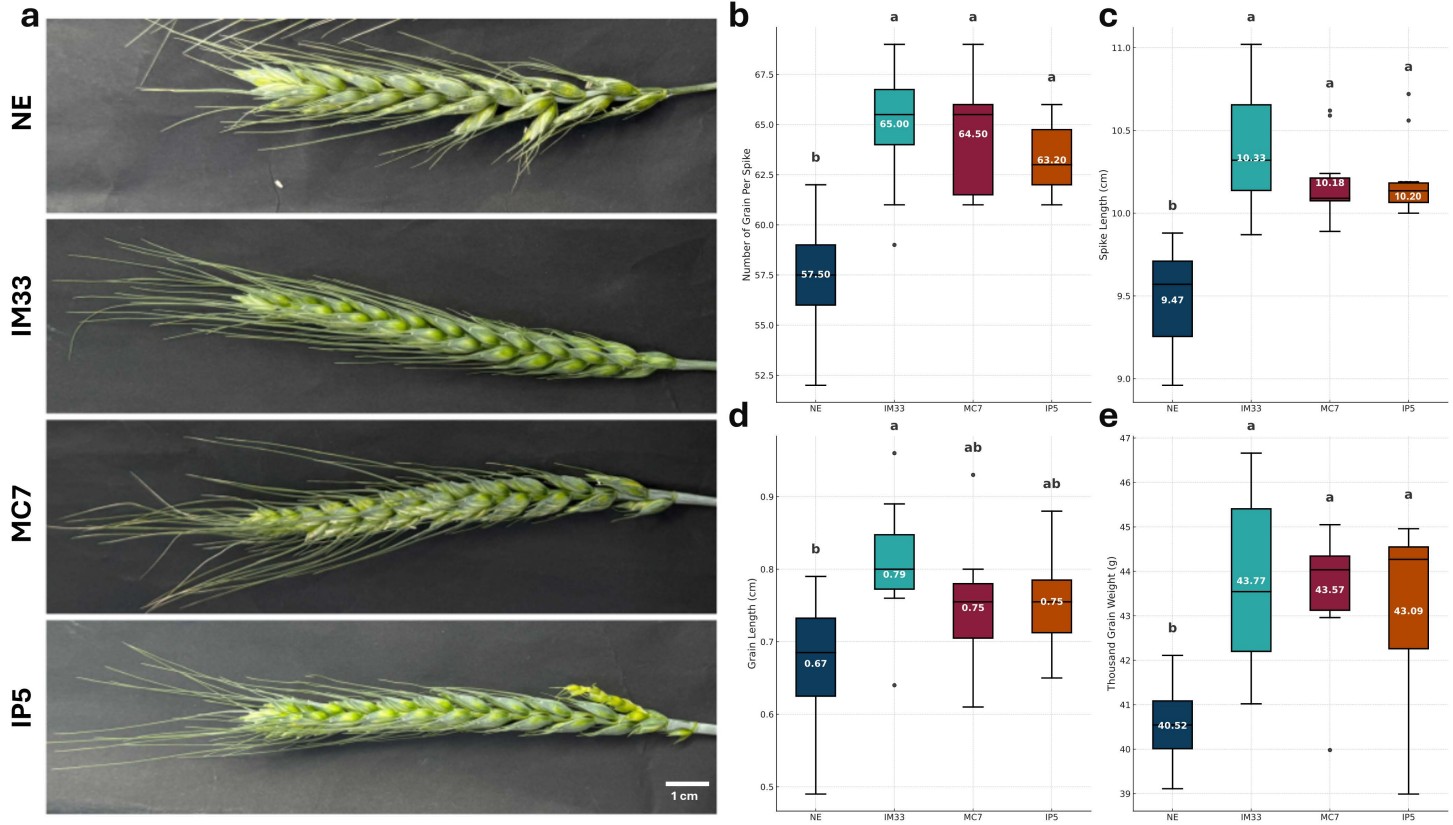

**Fig 4. Comparison of yield parameters among mutant lines generated by immature embryos, callus-derived, and *in planta* transformation, alongside the non-edited variety.** a) Representative spikes of mutant and non-edited plants are shown. Box plots illustrate differences in b) the number of grains per spike, c) spike length, d) grain length, and e) thousand-grain weight. The box plots were generated using ten measurements for each line, both non-edited and edited, for each phenotype. Statistical significance was determined by one-way ANOVA followed by Tukey's HSD test (*p* = 0.01). Bars sharing the same letter are not significantly different, while those with different letters differ significantly. All measurements are available in S6 Table.

to abiotic and biotic stress has become a central goal in modern breeding programs. Over the past decade, gene-editing technologies such as CRISPR/Cas9 have emerged as powerful tools to accelerate crop improvement. These approaches enable the precise modification of genomic sequences, allowing rapid introduction of desired traits into elite cultivars [2]. In wheat, the targeted knockout of negative regulators of yield, including *TaGW2* and *DUO-B1*, has been shown to enhance grain size, spikelet number, and yield [6,7].

Despite these advances, the widespread application of gene editing in wheat continues to be constrained by the challenges of stable transformation. The efficiency of established protocols varies widely depending on genotype, delivery strategy, and experimental parameters [13,28]. To address these limitations, a comparative optimization of multiple transformation strategies was undertaken in this study, including immature embryo transformation, callus-derived transformation from mature embryos, and injection-based *in planta* transformation. As a proof of concept, the *TaARE1-D* gene, previously identified as a negative regulator of nitrogen uptake and yield in rice and wheat [8,9] was targeted for knockout.

The transformation parameters were systematically adjusted to optimize efficiency for each method. Across all approaches, AGL1 consistently outperformed EHA105 and GV3101, confirming its suitability for wheat transformation (Fig 5a). Previous reports have attributed the high efficiency of AGL1 to the presence of the pSOUP helper plasmid, which is essential for the replication of pGreen-based vectors [29,30]. In this study, pSOUP-containing constructs were

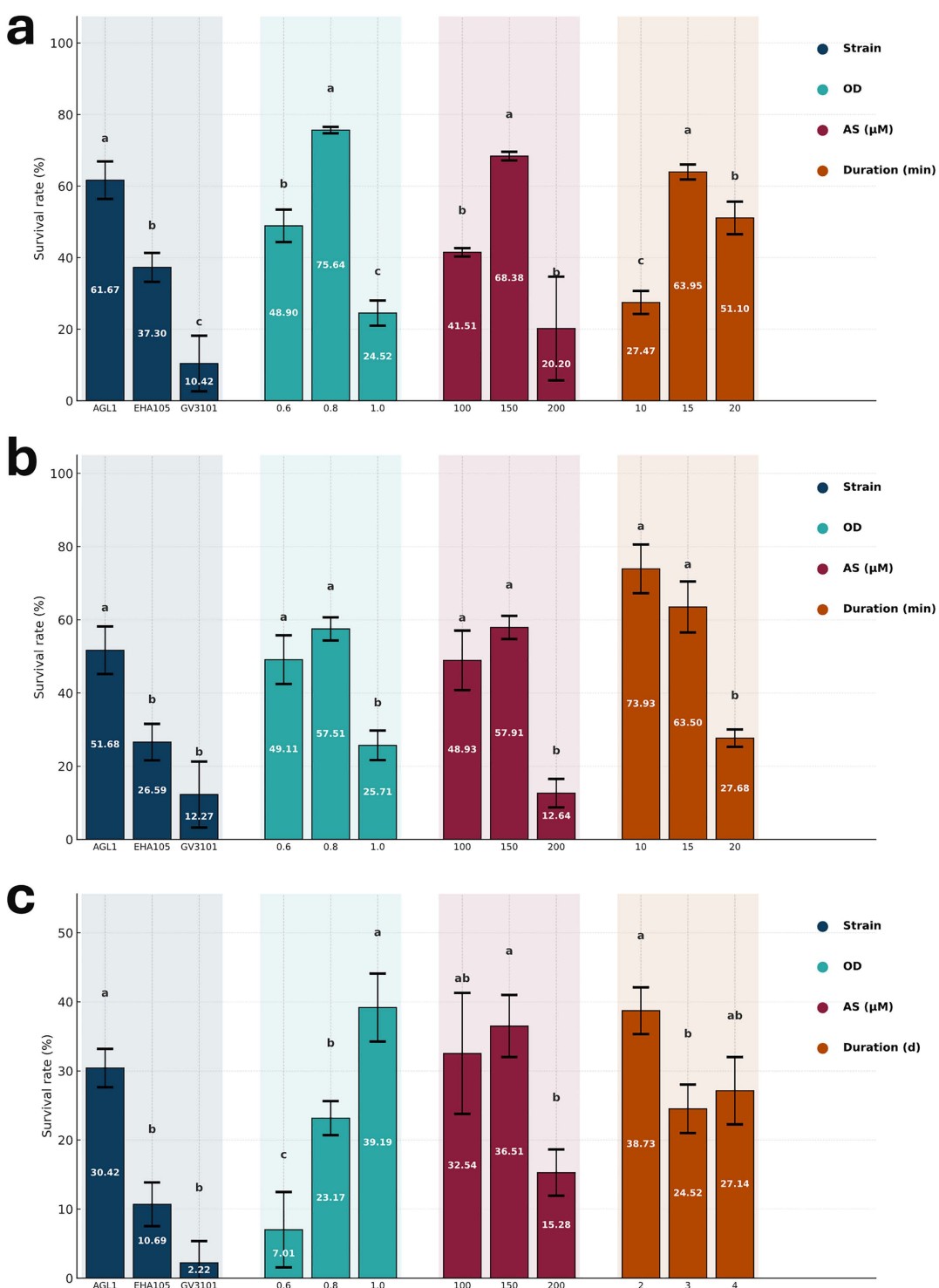

**Fig 5. Bar chart comparing survival and transformation efficiency across different methods.** a) represent hygromycin selection outcomes for immature embryo transformation depends on different parameters, b) for callus (from mature embryo) transformation, and c) show PCR screening results for *in planta* transformation. Statistical significance was determined at $p = 0.01$ (one-way ANOVA followed by Tukey's HSD test). Bar charts were generated using three replicates of each optimization parameter. The raw data is available in S7 Table.

introduced into all tested strains, suggesting that the superior performance of AGL1 may be linked to intrinsic compatibility with wheat tissues and more effective T-DNA delivery rather than solely the plasmid background.

After the effect of strain was established, the influence of bacterial density was assessed using AGL1. For *in vitro* transformation, an $OD_{600}$ of 0.8 proved optimal, as both lower (0.6) and higher (1.0) cell densities were associated with reduced survival of explants on hygromycin-containing medium (Fig 5). Lower densities are likely to reduce the frequency of T-DNA transfer events, while excessive bacterial loads at $OD_{600} = 1.0$ may impair explant viability through tissue damage or bacterial overgrowth [31]. Previous studies have reported successful immature embryo transformation at $OD_{600} = 0.5$ (13); however, the present findings suggest that efficiency can be improved by increasing the density to 0.8. In callus-derived transformation, no significant difference was observed between 0.6 and 0.8, although a reduction in survival was detected at $OD_{600} = 1.0$. In contrast, *in planta* transformation responded differently, as higher bacterial densities correlated with an increased frequency of PCR-positive plants. This divergence may be attributed to the absence of an enriched co-culture medium during *in planta* inoculation (Fig 5c). Without supplemental carbon sources such as glucose, lower bacterial densities may not support sufficient *Agrobacterium* activity for efficient T-DNA delivery, explaining the superior performance at OD600 = 1.0.

The concentration of acetosyringone in the inoculum and co-culture media was subsequently optimized. Among the tested concentrations, 150 µM consistently supported the highest transformation efficiency across all methods. In immature embryo transformation, this concentration produced a statistically significant improvement compared with both lower (100 µM) and higher (200 µM) levels (Fig 5a). In callus-derived and *in planta* transformation, the difference between 100 µM and 150 µM was not statistically significant, although a slight numerical advantage was observed at 150 µM (Fig 5c). By contrast, the use of 200 µM acetosyringone negatively affected transformation efficiency, suggesting potential inhibitory effects on either *Agrobacterium* activity or host tissue responses, as has been noted previously [32,33].

The effect of incubation time on transformation efficiency was also evaluated. In immature embryos, a 10-minute incubation with AGL1 was insufficient to promote high transformation efficiency, whereas extended exposure improved outcomes (Fig 5d). However, the response was found to be explant dependent. In callus-derived explants, prolonged incubation reduced survival on hygromycin-containing medium, likely due either to inefficient T-DNA transfer at later stages or to detrimental effects of extended bacterial contact on callus tissue integrity (Fig 5b). Previous reports have similarly noted that incubation periods of 10–12 minutes are optimal for callus transformation, whereas longer exposures compromise efficiency [21]. In *in planta* transformation, prolonged dark incubation following bacterial delivery reduced the number of PCR-positive plants (Fig 5c). The highest efficiency was obtained when seedlings were maintained for only two days, while extended incubation led not only to reduced transformation success but also to etiolation, particularly after four days in the dark.

Shoot regeneration from callus was further evaluated using 0.2 mg/L IAA and 1 mg/L zeatin. Callus from *T. aestivum* var. Kayra responded rapidly to IAA, with shoots appearing within 4–5 days, accompanied by simultaneous root initiation (S6 Fig). A similar response to IAA has been reported in callus derived from mature embryos of the Bobwhite S56 [21], although such rapid regeneration has not typically been described in callus from immature embryos. In both immature- and mature-derived callus, IAA proved more effective than zeatin, indicating that auxin supplementation promotes more efficient regeneration in this genotype. Although other concentrations of zeatin were not tested, the use of 1 mg/L was clearly less effective for Kayra regeneration.

Through optimization of transformation parameters, high efficiencies were achieved with immature embryo- and callus-derived explants, while lower but improved success was obtained with *in planta* transformation. When efficiencies were normalized using PCR screening to exclude escapes (Fig 6), the values remained comparable to previously reported protocols, confirming that the optimized approaches-maintained transformation quality. A major advance of this study was the modification of the immature embryo protocol. In earlier methods, embryos typically undergo three days

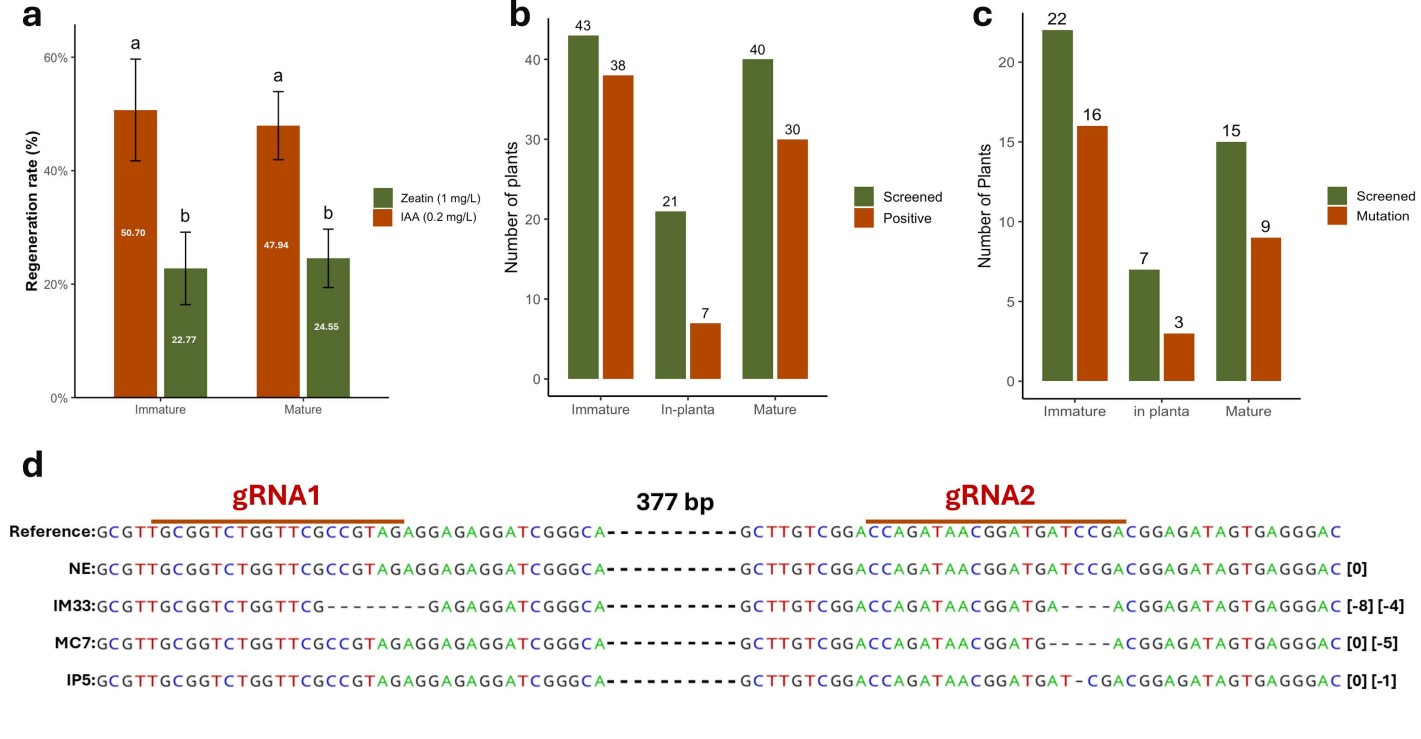

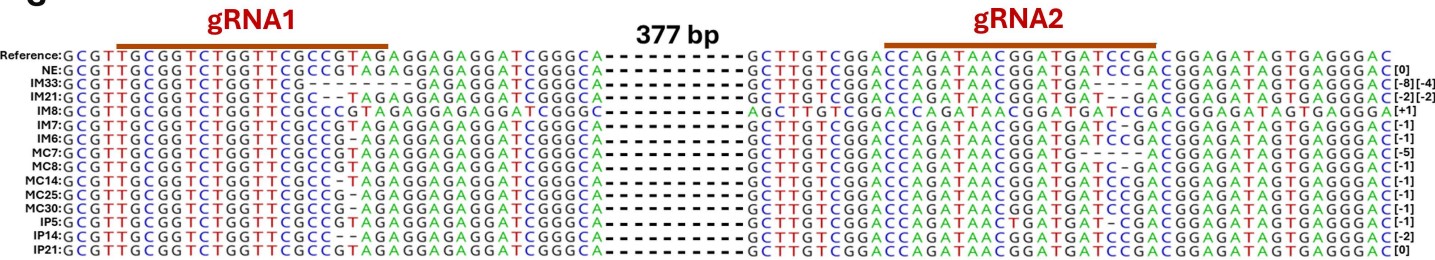

**Fig 6. Effects of hormones on regeneration efficiency and molecular screening of T-DNA integration and mutation types.** a) Bar graph comparing the effect of 0.2 mg/l IAA and 1 mg/l zeatin on regeneration from immature and mature embryo–derived calluses (three replicates) b) Number of plants screened and confirmed as transgenic using the optimized transformation methods. c) Number of plants carrying CRISPR/Cas9-induced mutations. Bars with the same letter are not significantly different from each other, while bars with different letters are significantly different according to Tukey's HSD ($p = 0.01$). The data is available in S8 Table) Mutation types identified in selected T0 plants. e) Mutation types identified in selected T1 plants.

of co-cultivation, a seven-day resting period, followed by a prolonged callus induction stage with selection lasting four to six weeks [13,14]. In contrast, embryos in this study were maintained for only two weeks on callus induction medium with selection before being transferred directly to regeneration medium containing IAA, under a 16/8-hour photoperiod that allowed regeneration and selection to proceed simultaneously. This adjustment shortened the overall transformation process by approximately one month. Although callus size was smaller and regenerated plantlets appeared less robust, efficiency remained high after normalization, indicating that the shortened induction period did not compromise transformation outcomes (S7 Fig). For callus transformation, the using of smaller calluses (6–10 mm) instead of the standard 15–20 mm reduced efficiency slightly but accelerated regeneration, and acclimatized plants showed no developmental abnormalities [13,21].

As discussed above, in previously described protocols for both immature and mature embryo transformation, researchers typically aimed to obtain well-developed callus. Although the formation of large, proliferating callus is generally considered a positive indicator of cellular competence, it requires additional subculturing steps before regeneration, thereby increasing labor demands. In contrast, the modified protocol developed in this study provides a continuous and easily manageable workflow, involving only the direct transfer of explants between media following embryo dissection. This streamlined approach effectively reduces workload and regeneration time while being well-suited for automation pipelines.

*In planta* transformation has recently gained attention as a promising alternative to conventional tissue culture–based approaches in cereal crops. Unlike traditional methods that rely on callus induction and regeneration, *in planta* systems enable direct gene delivery into developing meristems or embryonic tissues, offering a faster and less labor-intensive route for transgenic and genome-edited plant recovery. Recent studies have demonstrated that such approaches can substantially shorten transformation timelines and reduce somaclonal variation, making them increasingly attractive for species with strong genotype dependency [16]. Within this study, *in planta* optimization of inoculation parameters increased efficiency remarkably (3.3% to 33%) for *in planta* transformation compared with earlier reports from [22,23], demonstrating its potential as a rapid alternative to tissue culture-dependent approaches. This method eliminates the need for callus induction or regeneration. This reduces technical complexity and the time required to obtain edited lines (Table 1). While its efficiency was lower than other methods, our optimized *in planta* protocol is simpler and more scalable. Further improvements in infection uniformity and stable germline transmission are necessary to achieve performance comparable to that of optimized embryo-based methods.

CRISPR/Cas9-induced mutations were detected in plants generated by all three transformation methods. Although *in planta* transformation produced fewer overall transgenics, successful recovery of three *TaARE1-D* mutants demonstrated that the method could yield edited lines when optimized (Fig 6c and S8 Fig). The greater number of mutants obtained from immature embryo transformation reflected the higher number of transgenic plants produced, yet the proportion of mutants among tested plants was comparable between immature- and callus-derived approaches. On the other hand, CRISPR-induced mutations identified in IM33, MC7 and IP5 were reliably transmitted to the next generation (S9 Fig). In contrast,

**Table 1. Comparison of optimized wheat transformation methods for gene-editing across different aspects.**

| Parameter | Immature Embryo | Callus (Mature Embryo) | Injection (*In planta*) |
|---|---|---|---|
| **Transformation efficiency** | Highest (~66.8%) | High (~55.4%) | Moderate (~33.3%) |
| **Labor requirement** | Requires skilled labor | Requires skilled labor | Minimal skill required |
| **Cost/ Equipment** | Less affordable (tissue-culture media and sterile facilities) | Less affordable (tissue-culture media and sterile facilities) | More affordable (basic tools only) |
| **Time to obtain T$_1$ seeds** | ~220 days | ~160 days | ~90–100 days |
| **Somaclonal variation risk** | Higher | Higher | Very low |
| ***Genotype dependency** | High | High | Moderate (can vary by developmental stage) |
| **Scalability** | Moderate | Moderate | High |
| **Preferred application** | Precise, high-efficiency editing | Precise, high-efficiency editing | Rapid screening |
| **Major limitation** | Dependent on post-anthesis explant availability; long culture duration and labor-intensive process | Long culture time, labor-intensive | Lower stability; somatic, non-heritable edits |
| **Key advantage** | Highest editing accuracy and precise | Balanced efficiency and faster than immature embryo method | Simplest, fastest, and lowest cost |

*=The genotype dependency of *in planta* transformation was not experimentally tested in this study; however, this method is generally considered less genotype dependent [16].

no mutation was detected in the T1 progeny of IP21, even though the T0 plant tested positive by PCR screening (Fig 6e). The absence of mutations in the T1 progeny of IP21 suggests that the mutation was somatic rather than germline. Meanwhile, the IP14 and IP5 lines, which were also PCR-positive at the T0 stage, produced T1 plants with one- and two-base-pair deletions, respectively, confirming true heritable mutations (S1 Appendix). It has been shown that the *in-planta* method requires improvement, especially for targeting the wheat apical meristem more precisely. This refinement would reduce mutations restricted to somatic tissues and decrease chimerism, thereby increasing the likelihood of heritable edits.

Phenotypic evaluation confirmed that *TaARE1-D* loss-of-function enhanced yield-related traits, consistent with previous studies in wheat and rice [8,9]. Increases in spike length, grain number, and thousand-grain weight were observed in mutants generated by all transformation methods, including *in planta*. Grain length varied among lines, with IP5 (*in planta*) and MC7 (callus-derived) grouping statistically with both the non-edited control and the immature embryo mutant IM33, likely due to variance within those lines. For other parameters, all mutants exhibited significant improvements compared with the non-edited control. In addition, the stay-green phenotype that is a hallmark of the *TaARE1-D* disruption was consistently observed, confirming the functional knockout.

The protocols optimized here improved both transformation success and the time required for recovery of edited lines. These improvements provide faster, more reliable platforms for CRISPR/Cas9-mediated genome editing and enable the development of high-yield, stress-tolerant wheat varieties. Although the optimized transformation methods demonstrated high efficiency and reproducibility, the experiments were conducted using only one wheat genotype, *T. aestivum* cv. Kayra. This genotype-specific focus is a limitation because transformation and regeneration responses in wheat vary considerably among cultivars due to differences in tissue physiology and *Agrobacterium* susceptibility. Future work should therefore validate and refine these optimized parameters, such as bacterial density, acetosyringone concentration, and incubation duration, across a broader range of genotypes, including elite and recalcitrant cultivars. Despite this limitation, the systematic optimization framework established here provides a practical foundation for integrating CRISPR/Cas9-mediated genome editing into modern breeding programs. Using these improved transformation systems with high-yielding or stress-resilient varieties will speed up functional genomics studies and help develop next-generation wheat germplasm with greater productivity and resilience in changing climates.

## Supporting information

**S1 Fig. Schematic representation of the TaARE1-D gene structure in Triticum aestivum and the CRISPR/Cas9 constructs used for genome editing.**
(JPG)

**S2 Fig. Cloning and verification of CRISPR/Cas9 plasmids targeting TaARE1-D.**
(JPG)

**S3 Fig. Over-tillering phenotype observed in wheat plants transformed by the in planta method three weeks after Agrobacterium injection.**
(JPG)

**S4 Fig. Transgene screening of regenerated wheat plants derived from immature embryo, mature embryo callus, and in planta transformations following hygromycin selection.**
(TIF)

**S5 Fig. Screening of regenerated transgenic wheat plants using ACT-PCR.** Absence of amplification indicates CRISPR/Cas9-induced mutations at the gRNA1 target site.
(TIF)

**S6 Fig. Callus after Agrobacterium-mediated transformation and regenerated calli after transfer to IAA-containing selection medium.**
(JPG)

**S7 Fig. Regenerated calli derived from transformed immature embryos and regenerated plants after two months on selection medium.**
(JPG)

**S8 Fig. Spike morphology, stay-green phenotype, harvested spikes, and seeds of edited and non-edited wheat lines.**
(JPG)

**S9 Fig. Sequence chromatogram alignment of TaARE1-D from non-edited and edited lines with gRNA target sites highlighted.**
(JPG)

**S1 Table. Sequences of gRNAs and primers used in this study.**
(XLSX)

**S2 Table. Immature embryo transformation efficiency under hygromycin selection across different Agrobacterium strains and experimental parameters.**
(XLSX)

**S3 Table. Transformation efficiency of callus derived from mature embryos under hygromycin selection.**
(XLSX)

**S4 Table. Comparison of regeneration efficiency between IAA and zeatin in immature and mature embryo-derived calli.**
(XLSX)

**S5 Table. In planta transformation efficiency under hygromycin selection.**
(XLSX)

**S6 Table. Phenotypic traits of edited and non-edited wheat lines.**
(XLSX)

**S7 Table. Raw data for transformation optimization experiments.**
(XLSX)

**S8 Table. Raw data for IAA and zeatin effects on regeneration efficiency.**
(XLSX)

**S9 Table. Genotype and mutation profiles of phenotyped CRISPR/Cas9-edited wheat lines.**
(XLSX)

**S1 Appendix. Mutations in generated mutant lines.**
(DOCX)

## Author contributions

**Conceptualization:** Mumin Ibrahim Tek, HAKAN FIDAN.

**Data curation:** Mumin Ibrahim Tek, Kubra Budak Tek, Pelin Sarikaya.

**Formal analysis:** Mumin Ibrahim Tek, Kubra Budak Tek, Pelin Sarikaya, Abdul Razak Ahmed.

**Methodology:** Mumin Ibrahim Tek, Kubra Budak Tek, HAKAN FIDAN.

**Project administration:** HAKAN FIDAN.

**Supervision:** HAKAN FIDAN.

**Validation:** Mumin Ibrahim Tek.

**Visualization:** Kubra Budak Tek.

**Writing – original draft:** Mumin Ibrahim Tek, HAKAN FIDAN.

**Writing – review & editing:** Kubra Budak Tek, HAKAN FIDAN.

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
