## [Decision Letter · Decision Letter 0]

22 Oct 2025

Dear Dr. FIDAN,

Thank you for submitting your manuscript to PLOS ONE. After careful consideration, we feel that it has merit but does not fully meet PLOS ONE’s publication criteria as it currently stands. Therefore, we invite you to submit a revised version of the manuscript that addresses the points raised during the review process.

Specifically, your manuscript was reviewed by two independent experts in the field. Both reviewers find the work interesting but raised several issues which need to be addressed properly. The reviewers provide detailed comments in their reviews and pointed out the areas where the manuscript needs to be improved. Therefore, manuscript needs a major revision at this time to meet the publication standard of PLOS ONE.

We look forward to receiving your revised manuscript.

Kind regards,

Mohammad Irfan, Ph.D.

Academic Editor

PLOS ONE

Journal Requirements:

4. Please remove all personal information, ensure that the data shared are in accordance with participant consent, and re-upload a fully anonymized data set.

Reviewers' comments:

Reviewer's Responses to Questions

**Comments to the Author**

1. Is the manuscript technically sound, and do the data support the conclusions?

Reviewer #1: No

Reviewer #2: Yes

2. Has the statistical analysis been performed appropriately and rigorously?

Reviewer #1: No

Reviewer #2: Yes

3. Have the authors made all data underlying the findings in their manuscript fully available?

Reviewer #1: Yes

Reviewer #2: Yes

4. Is the manuscript presented in an intelligible fashion and written in standard English?

Reviewer #1: Yes

Reviewer #2: Yes

Reviewer #1: Review Comments

The manuscript " Choosing the Best Route: Comparative Optimization of Wheat Transformation Methods for Improving Yield by Targeting TaARE1-D with CRISPR/Cas9" by Tek et al. is a good protocol optimisation of genome editing in wheat. This research article provides importance of transformation tissues for genome editing efficiencies. This manuscript is of interest to the crop biology researchers, as well as different biology researchers and I expect that the article will be well-cited, but it lacks presentation. I have the following comments to consider.

1. Authors provide high resolution figures and graphs. Graphs should have data points

2. Authors should provide all the raw data in the supplementary files and add labels in the tables.

3. Figure and table legends should include all the details (biological, technical ,statistical, etc) parameters.

4. Authors standardize all the materials and methods or they have taken from any literature. If taken, cite all the articles in the text.

5. Authors should add scale bar in all the images.

6. Wheat is polyploidy species, So authors should provide the genotypic data of all the generations till the generation of phenotype measured in a tabulated manner for better understanding of homozygous and hemizygous mutants of the TaARE1-D, and how they have selected the knockout mutant plants without marker.

Reviewer #2: Authors presented a comparative optimization of three wheat transformation methods: immature embryo, callus-derived, and in planta transformation to improve efficiency and reduce regeneration time for CRISPR/Cas9-based genome editing. The study is well-structured, and the authors have attempted to address a key limitation in wheat breeding and gene editing, namely, low transformation efficiency. The experimental design is comprehensive, and the results are promising, demonstrating substantial improvement in transformation efficiency and successful validation using TaARE1-D knockout. However, there are several points that need clarification, additional validation, and more critical discussion before the manuscript can be considered for publication.

1. The study evaluates transformation efficiency using only a single wheat genotype (cv. Kayra). Given that wheat transformation is highly genotype-dependent, the applicability of the optimized protocols across diverse cultivars remains uncertain. The authors should acknowledge this limitation more clearly in the Discussion and, if possible, provide preliminary data indicating whether similar improvements can be expected in other genotypes.

2. There is no molecular confirmation for example, sequencing of edited loci or off-target analysis for the CRISPR-induced mutations. PCR absence alone is not a reliable indicator of editing.

3. Are the authors sure about the heritability and stability of the edited traits across T1 or T2 generations? This information is essential to confirm stable transmission of edits.

4. It would be nice to have a summary table comparing different aspects like time, cost, efficiency, genotype dependence.

5. The in planta transformation method involves directly injecting Agrobacterium carrying the CRISPR/Cas9 construct into young seedlings (usually the apical meristem). This approach is appealing because it avoids tissue culture, but it is also less predictable. The in planta transformation method, while faster, shows notably lower efficiency and lacks clarity on whether transformation occurred in somatic or germline cells. This distinction is critical, as somatic edits may not be heritable. The authors should provide evidence of edit transmission to progeny (sequencing data).

6. Clearly state limitations and future applications, such as use in elite cultivars or integration into breeding programs.

7. It is very important to include the details on plant growth conditions during regeneration e.g., light intensity, humidity.

8. L127-128: Excess bacterial suspension was removed by blotting 128 on sterile filter paper (Supplementary Figure 3). This part refers to Sup. Fig. 3 to show filter paper blotting of excess bacteria, whereas Sup. Fig. 3 is about another thing, it is a mixture of photos both for immature and mature embryos transformation (no pictures of filter paper are seen here). It is better to remove this reference to sup. fig? It looks so confusing at this part of the paper.

9. This section of the paper "Transformation of Callus derived from Mature Embryos and Regeneration" is represented in figure 2 of the paper (not supplementary), but it is not mentioned in the text that is figure 2. Also, figure 2 starts with transformed callus going to regeneration and later stages. It is better to move mature seeds callus induction from supplementary to here to show the complete process. Also, this way that Sup. fig. 3 will not be a mixture of two immature and mature.

10. L148-150: “Embryos were 149 incubated for 10, 15, or 20 minutes, with tubes inverted three to four times every five minutes to 150 ensure uniform exposure (Supplementary Figure 3)”. Sup. fig. 3 only a&b are about this process. It is better to include it in figure 1, this immature embryo transformation part is shown in figure 1 of the paper but it doesn't state so. Direction to the figure is better to put as a whole to a process or step by step (1a, 1b, etc), but not dividing it into two, one in main figure, one in supplementary figure, and the sentence it is attached to "inverted tubes 3-4 times" is not actually shown on photos.

11. L198-199: “Transformation efficiency was assessed based on the survival of 199 explants following callus formation and selection with 15 mg/L hygromycin (Figure 1c).” This part is about comparison of different parameters (OD, AS concentration) but mentions Figure 1c, which is just a picture of one plate. I think this part of the results is better represented in supplementary table 2, it is worth referring to it, not figure 1c.

12. L208: Legend of Figure 1 is incomplete

13. Supplementary figure 3 is a mixture of immature embryo process (a&b) and mature embryo callus induction (c-e). It is better to either separate it into two different figures, one for the immature, another for the mature. Or include them into the main figures of the paper.

14. The discussion needs a more critical comparison with previous wheat transformation studies (e.g., Hayta et al., 2019; Ye et al., 2023).

15. In discussion, the authors should address genotype dependence, the experiments were conducted using only cv. Kayra, which is the limitation.

The manuscript presents a potentially impactful methodological advance in wheat transformation. However, it requires additional experimental evidence (especially molecular validation of edits especially for in planta transformation method and broader discussion) to substantiate its claims. Addressing the above issues would considerably enhance the scientific robustness and clarity of the study.

**Do you want your identity to be public for this peer review?** For information about this choice, including consent withdrawal, please see our Privacy Policy

Reviewer #1: No

Reviewer #2: **Yes:**  Urooj Fatima

---

## [Author Response · Author response to Decision Letter 1]

4 Dec 2025

Reviewer 1

Comment) The manuscript " Choosing the Best Route: Comparative Optimization of Wheat Transformation Methods for Improving Yield by Targeting TaARE1-D with CRISPR/Cas9" by Tek et al. is a good protocol optimisation of genome editing in wheat. This research article provides importance of transformation tissues for genome editing efficiencies. This manuscript is of interest to the crop biology researchers, as well as different biology researchers and I expect that the article will be well-cited, but it lacks presentation.

Response: Thank you so much for considering reviewing our paper. Here are the revisions according to your recommendations.

Comment 1) I have the following comments to consider. Authors provide high resolution figures and graphs. Graphs should have data points

Response: Data values are added into the bars in Figure 6a. Figure 4 and 5 are reconstructed for better visualization with data values.

Comment 2) Authors should provide all the raw data in the supplementary files and add labels in the tables.

Response: The raw data for each replicate of the transformation optimization and the effects of indole-3-acetic acid (IAA) and zeatin are presented in Supplementary Tables 7 and 8 of the supplementary document and are labeled.

Comment 3) Figure and table legends should include all the details (biological, technical ,statistical, etc) parameters.

Response: The method of comparison is incorrectly stated in the Methods section and has been corrected as;

“One-way ANOVA was used to compare mean trait values, followed by Tukey HSD for post hoc comparisons”.

The table and graphics are annotated with all requested details.

Comment 4) Authors standardize all the materials and methods or they have taken from any literature. If taken, cite all the articles in the text.

Response: Although it is cited in the discussion (22, 23), we forgot to cite it in the Materials and Methods section due to a mistake regarding in-planta transformation. It has been clearly described in the Methods and Materials section with citation (L162-163).

“The in planta transformation method has been adapted and optimized from the previously indicated methods (22,23).”

Comment 5) Authors should add scale bar in all the images.

Response: Added for both figure in manuscript (Figure 1, 2, and 3) and supplementary figures (Sup Fig 3, 6, 7, and 8).

Comment 6) Wheat is polyploidy species, So authors should provide the genotypic data of all the generations till the generation of phenotype measured in a tabulated manner for better understanding of homozygous and hemizygous mutants of the TaARE1-D, and how they have selected the knockout mutant plants without marker.

Response: Thank you very much for this helpful suggestion. ACT-PCR was selected because it provides a rapid and reliable approach for identifying homozygous mutants, although it does not distinguish hemizygous genotypes. Our focus on homozygous mutants was based on previous work (Zhang et al., 2021) showing that homozygous taare1d lines exhibit improved yield-related traits. Following the reviewer’s recommendation, we have now clarified this method in the Materials and Methods section, including the use of D-genome–specific primers designed to selectively amplify TaARE1-D (L184–188).

In addition, we performed Sanger sequencing to validate the mutations identified using the newly designed TaARE1-D–specific primers. We included both transgenic-positive T0 plants and their corresponding T1 individuals in order to detect potential heterozygous or chimeric edits that could not be resolved by ACT-PCR alone. This procedure is now described in detail in the Materials and Methods section (L191–195).

Finally, the Results and Discussion sections have been expanded to include the Sanger sequencing data that confirm the heritability of the induced mutations.

Reviewer 2

Comment) Authors presented a comparative optimization of three wheat transformation methods: immature embryo, callus-derived, and in planta transformation to improve efficiency and reduce regeneration time for CRISPR/Cas9-based genome editing. The study is well-structured, and the authors have attempted to address a key limitation in wheat breeding and gene editing, namely, low transformation efficiency. The experimental design is comprehensive, and the results are promising, demonstrating substantial improvement in transformation efficiency and successful validation using TaARE1-D knockout. However, there are several points that need clarification, additional validation, and more critical discussion before the manuscript can be considered for publication.

Response: Thank you so much for considering reviewing our paper. Here are the revisions according to your recommendations.

Comment 1) The study evaluates transformation efficiency using only a single wheat genotype (cv. Kayra). Given that wheat transformation is highly genotype-dependent, the applicability of the optimized protocols across diverse cultivars remains uncertain. The authors should acknowledge this limitation more clearly in the Discussion and, if possible, provide preliminary data indicating whether similar improvements can be expected in other genotypes.

Response: Thank you for this valuable recommendation. We have revised and expanded the closing section of the Discussion to clearly acknowledge the genotype dependency of wheat transformation and the limitation of using a single genotype (T. aestivum cv. Kayra). The revised text discusses the potential variability among cultivars and emphasizes the need for future validation of the optimized parameters across diverse and elite genotypes.

Comment 2) There is no molecular confirmation for example, sequencing of edited loci or off-target analysis for the CRISPR-induced mutations. PCR absence alone is not a reliable indicator of editing.

Response: Molecular confirmation of the CRISPR-induced mutations was performed by Sanger sequencing of both T0 and T1 plants. The sequencing procedure has now been fully described in the Materials and Methods section. The corresponding results have been added to the Results and Discussion sections, where the edited loci are presented and discussed. The raw chromatograms are provided in Supplementary Figure 9, and the sequence alignments are included within the main manuscript (Figure 6d-6e) to clearly show the detected mutations.

Comment 3) Are the authors sure about the heritability and stability of the edited traits across T1 or T2 generations? This information is essential to confirm stable transmission of edits.

Response:. We examined the heritability of the CRISPR-induced mutations in the T1 generation derived from edited T0 plants. Sanger sequencing of the IM33, MC7, and IP5 lines confirmed that the same mutations present in the T0 plants were transmitted to their T1 progeny, demonstrating stable inheritance of the edits. These results are clearly presented in both the Results and Discussion sections and are shown in Figure 6.

Comment 4) It would be nice to have a summary table comparing different aspects like time, cost, efficiency, genotype dependence.

Response: According to the recommendations, Table 1, which compares methods in terms of different aspects, has been included in the manuscript.

Comment 5) The in planta transformation method involves directly injecting Agrobacterium carrying the CRISPR/Cas9 construct into young seedlings (usually the apical meristem). This approach is appealing because it avoids tissue culture, but it is also less predictable. The in planta transformation method, while faster, shows notably lower efficiency and lacks clarity on whether transformation occurred in somatic or germline cells. This distinction is critical, as somatic edits may not be heritable. The authors should provide evidence of edit transmission to progeny (sequencing data).

Response: Thank you very much for this constructive comment. In accordance with the reviewer’s suggestion, we performed Sanger sequencing on both T0 and T1 plants generated through the in-planta transformation method. Our results show that the T1 progeny of the IP5 line carried the same one-base-pair deletion at the gRNA2 site that was detected in its T0 plant, confirming successful transmission of the edit to the next generation. In contrast, no mutation was identified in the T1 plants of IP21, despite its T0 plant being PCR-positive, suggesting that the edit in IP21 was somatic and therefore not heritable. This interpretation is further supported by the IP14 line, which was also PCR-positive at the T0 stage and produced T1 progeny with a two-base-pair deletion. These findings have been incorporated into the Results and Discussion sections, where the heritability and variability of edits obtained through the in-planta method are discussed.

Comment 6) Clearly state limitations and future applications, such as use in elite cultivars or integration into breeding programs.

Response: The Discussion section has been revised to clearly state the study’s limitations and future applications. The added paragraph emphasizes the potential of applying the optimized transformation systems to elite and stress-tolerant wheat cultivars and discusses their integration into modern breeding programs to accelerate genome editing–based crop improvement.

Comment 7) It is very important to include the details on plant growth conditions during regeneration e.g., light intensity, humidity.

Response: Added (500-550 μmol m−2 s−1) and 70% humidity in M&M section.

Comment 8) L127-128: Excess bacterial suspension was removed by blotting 128 on sterile filter paper (Supplementary Figure 3). This part refers to Sup. Fig. 3 to show filter paper blotting of excess bacteria, whereas Sup. Fig. 3 is about another thing, it is a mixture of photos both for immature and mature embryos transformation (no pictures of filter paper are seen here). It is better to remove this reference to sup. fig? It looks so confusing at this part of the paper.

Response: Sorry for the confusion. It has been removed.

Comment 9) This section of the paper "Transformation of Callus derived from Mature Embryos and Regeneration" is represented in figure 2 of the paper (not supplementary), but it is not mentioned in the text that is figure 2. Also, figure 2 starts with transformed callus going to regeneration and later stages. It is better to move mature seeds callus induction from supplementary to here to show the complete process. Also, this way that Sup. fig. 3 will not be a mixture of two immature and mature.

Response: The Supplementary Figure 3 has been removed, and the image associated with the mature embryo transformation has been combined with Figure 2 in the manuscript. The caption of Figure 2 has also been revised.

Comment 10) L148-150: “Embryos were 149 incubated for 10, 15, or 20 minutes, with tubes inverted three to four times every five minutes to 150 ensure uniform exposure (Supplementary Figure 3)”. Sup. fig. 3 only a&b are about this process. It is better to include it in figure 1, this immature embryo transformation part is shown in figure 1 of the paper but it doesn't state so. Direction to the figure is better to put as a whole to a process or step by step (1a, 1b, etc), but not dividing it into two, one in main figure, one in supplementary figure, and the sentence it is attached to "inverted tubes 3-4 times" is not actually shown on photos.

Response: Supplementary Fig. 3 has been removed, and the image associated with collecting immature embryos and transformation has been incorporated into Figure 1. The caption has been revised. The manuscript has also been revised for the reference figures.

Comment 11) L198-199: “Transformation efficiency was assessed based on the survival of 199 explants following callus formation and selection with 15 mg/L hygromycin (Figure 1c).” This part is about comparison of different parameters (OD, AS concentration) but mentions Figure 1c, which is just a picture of one plate. I think this part of the results is better represented in supplementary table 2, it is worth referring to it, not figure 1c.

Response: Revised.

Comment 12) L208: Legend of Figure 1 is incomplete

Response: Completed.

Comment 13) Supplementary figure 3 is a mixture of immature embryo process (a&b) and mature embryo callus induction (c-e). It is better to either separate it into two different figures, one for the immature, another for the mature. Or include them into the main figures of the paper.

Response: Supplementary Figure 3 has been removed, and the associated images have been incorporated into Figures 1 and 2, which are associated with immature embryo dissection/transformation, and mature embryo transformation, respectively.

Comment 14) The discussion needs a more critical comparison with previous wheat transformation studies (e.g., Hayta et al., 2019; Ye et al., 2023).

Response: A more comparative discussion of all transformation methods can be found in the discussion section (L444-466). In addition to required time, we compare the necessary labor and technical skills. Considering its lower transformation efficiency, we extended the in-plant transformation method section with its advantages.

Comment 15) In discussion, the authors should address genotype dependence, the experiments were conducted using only cv. Kayra, which is the limitation.

Response: A paragraph discussing genotype dependency and acknowledging the limitation of using a single genotype (T. aestivum cv. Kayra) has been added to the last paragraph of the discussion section.

---

## [Decision Letter · Decision Letter 1]

6 Jan 2026

Dear Dr. FIDAN,

Thank you for submitting your manuscript to PLOS ONE. After careful consideration, we feel that it has merit but does not fully meet PLOS ONE’s publication criteria as it currently stands. Therefore, we invite you to submit a revised version of the manuscript that addresses the points raised during the review process.

Thank you for revising the manuscript, however reviewer 1 has a concern regarding the selection. Please address that and submit the revised version.

We look forward to receiving your revised manuscript.

Kind regards,

Mohammad Irfan, Ph.D.

Academic Editor

PLOS One

Journal Requirements:

Reviewers' comments:

Reviewer's Responses to Questions

**Comments to the Author**

Reviewer #1: (No Response)

2. Is the manuscript technically sound, and do the data support the conclusions?

Reviewer #1: Partly

3. Has the statistical analysis been performed appropriately and rigorously?

Reviewer #1: Yes

4. Have the authors made all data underlying the findings in their manuscript fully available?

Reviewer #1: No

5. Is the manuscript presented in an intelligible fashion and written in standard English?

Reviewer #1: Yes

Reviewer #1: Authors have not addressed my last review question: Wheat is polyploidy species, so authors should provide the genotypic data of all the generations till the generation of phenotype measured in a tabulated manner for better understanding of homozygous and hemizygous mutants of the TaARE1-D, and how they have selected the knockout mutant plants without marker.

**Do you want your identity to be public for this peer review?** For information about this choice, including consent withdrawal, please see our Privacy Policy

Reviewer #1: No

---

## [Author Response · Author response to Decision Letter 2]

18 Jan 2026

Comment) Authors have not addressed my last review question: Wheat is polyploidy species, so authors should provide the genotypic data of all the generations till the generation of phenotype measured in a tabulated manner for better understanding of homozygous and hemizygous mutants of the TaARE1-D, and how they have selected the knockout mutant plants without marker.

Response: This point was unintentionally overlooked during the previous revision. We have now added a new table (Supplementary Table 9) summarizing the genotypic characterization of the IM33, MC7, and IP5 lines across generations up to the phenotyped generation. In addition, the Phenotyping and Data Analysis section of the Materials and Methods has been expanded to clarify the selection strategy for knockout lines, as follows:

“Edited knockout lines were selected based on the effects of the induced mutations on the TaARE1-D protein sequence. The mutations were interpreted by comparing the predicted amino acid sequences with that of the reference TaARE1-D protein (TraesCS7D03G0650400). Only plants carrying frameshift mutations or premature stop codons were advanced for phenotypic analysis.”

---

## [Editor Report · Decision Letter 2]

25 Jan 2026

Choosing the Best Route: Comparative Optimization of Wheat Transformation Methods for Improving Yield by Targeting TaARE1-D with CRISPR/Cas9

PONE-D-25-50562R2

Dear Dr. FIDAN,

We’re pleased to inform you that your manuscript has been judged scientifically suitable for publication and will be formally accepted for publication once it meets all outstanding technical requirements.

Kind regards,

Mohammad Irfan, Ph.D.

Academic Editor

PLOS One
---

## [Editor Report · Acceptance letter]

PONE-D-25-50562R2

PLOS One

Dear Dr. FIDAN,

I'm pleased to inform you that your manuscript has been deemed suitable for publication in PLOS One. Congratulations! Your manuscript is now being handed over to our production team.

Kind regards,

on behalf of

Dr. Mohammad Irfan

Academic Editor

PLOS One